# Investigating immune amnesia after measles virus infection in two West African countries: A study protocol

Karine Fouth Tchos[1], Renée Ridzon[1], Mory Cherif Haidara[2], Djeneba Dabitao[3], Esther Akpa[1], Daouda Camara[2], David Vallée[4], Mariam Coulibaly[3], Sekou Camara[2], Jamila Aboulhab[1], Mahamadou Diakité[3], Bassirou Diarra[3], Samba Diarra[3], Ilo Dicko[3], Alyson Francis[5], Cécé Francis Kolié[2], Michel Koropogui[2], Caeul Lim[4], Seydou Samaké[3], Sally Hunsberger[1]*, Moussa Sidibé[2], Ray Y. Chen[1], Issa Konate[3], Seydou Doumbia[3], Abdoul Habib Beavogui[2], Kathryn Shaw-Saliba[1]

**1** Division of Clinical Research, NIAID/NIH, Rockville, Maryland, United States of America, **2** Centre National de Formation et de Recherche en Santé Rurale (CNFRSR) de Maferinyah, Conkary, Guinea, **3** University Clinical Research Center, University of Sciences, Techniques and Technologies of Bamako, Bamako, Mali, **4** Clinical Monitoring Research Program Directorate (CMRPD), Frederick National Laboratory for Cancer Research, Frederick, Maryland, United States of America, **5** Systex, Inc, Rockville, Maryland, United States of America

* sally.hunsberger@nih.gov

## Abstract

"Investigation of Immune Amnesia Following Measles Infection in Select African Regions" (ClinicalTrials.gov Identifier: NCT06153979) is a prospective, observational, longitudinal study being conducted in two West African countries; Guinea and Mali. Acute measles virus (MeV) infection has been shown to result in a loss of pre-existing immunity (immune amnesia). MeV-induced immune amnesia has not been studied in West Africa where continual MeV outbreaks occur. Additionally previous studies have relied on naturally occurring exposures to viruses to examine the immune systems ability to create antibodies. Thus, the overall goal of this protocol is to investigate the impact of MeV infection on pre-existing immunity to endemic pathogens in West Africa, observe the effect of a subsequent exposure to a novel pathogen (rabies vaccine), and measure the frequency of subsequent healthcare visits. A total of 256 children aged 1–15 years are being enrolled into one of two study arms: those with acute MeV infection (cases) as confirmed by laboratory testing and without (controls). Controls must be immune to MeV (have IgG). Blood samples are collected at multiple time points including screening (Day 0), at an optional visit to repeat IgM serology for inconclusive or negative Day 0 results (Day 7–10), and during follow-up visits on Day 14, Week 13, and Week 52. These blood samples will be tested to evaluate both humoral and cellular immune responses to a panel of viruses, bacteria, and parasites, including pathogens endemic to West Africa. To explore how recent MeV infection may affect the child's ability to respond to a new exposure, all participants will receive a rabies vaccine (as a controlled

**Data availability statement:** This article does not report any data and the data availability policy is not applicable.

**Funding:** The author(s) received no specific funding for this work.

**Competing interests:** The authors have declared that no competing interests exist.

stimulus) at one of two timepoints post Day 0 visit. Biological samples will be collected after vaccination to assess if the rabies vaccine response differs: 1) between cases and controls, and 2) based on the time since acute MeV infection. In addition, the study team will collect information on healthcare encounters during the year-long follow-up to determine if there is a difference in the number of encounters by study group. The findings of this study will further the understanding of the MeV immune amnesia phenotype by understanding its impact on endemic pathogens and subsequent immune response following infection.

## Introduction

Measles virus (MeV) infection is a major cause of death globally for children under five years of age [1–3]. There are reports of increases in cases in many countries, including in West Africa [4,5]. The Expanded Program on Immunization in Guinea and Mali calls for two doses of measles vaccine starting at 9 months [6,7]. According to WHO/UNICEF national immunization coverage estimates for the African region in 2022, 69% of children received the first measles vaccine and 45% received the second [6]. The rates in Mali were very close to these averages, at 70% for the first dose and 44% for the second dose, but Guinea lagged farther behind, at 47% and 3%, respectively [8]. These rates are far below the 92–95% level required for herd immunity and control of community transmission of measles [9]. This is primarily because, over the past decade, West Africa has experienced measles outbreaks following large-scale public health emergencies, such as the Ebola epidemic and the COVID-19 pandemic, which have disrupted global routine childhood vaccine coverage [10–12]. Armed conflict has also disrupted routine vaccinations in many countries, resulting in a disproportionate number of cases of vaccine preventable diseases, including measles [13,14]. Finally, some vaccination campaigns may have used vaccines where appropriate storage conditions were not maintained due to a lack of available resources or equipment. Thus, the presence of global measles outbreaks underscores the importance of studies that will contribute to a better understanding of the effect of MeV infection on morbidity and mortality.

MeV infection not only directly causes morbidity but is also associated with increased susceptibility to other infections and/or increased hospitalizations [15–19]. Measles vaccination has been associated with larger reductions in morbidity and mortality than would be expected if caused by acute MeV infection alone [19–24]. While there is a strong immune response to MeV infection that confers life-long immunity to MeV there is some evidence of a loss of existing immunity to other pathogens. This phenomenon, known as "immune amnesia" [25], is associated with a loss of memory B cells and reduced antibody repertoires from previous infections and vaccinations [26,27]. MeV-induced immune amnesia has been characterized in animal studies with ferrets and rhesus macaques and has also been examined in European populations [26,28,29]. Given the current outbreaks of measles in West Africa, this study aims to determine whether measles-induced immune amnesia occurs in

African regions and affects immune responses to endemic pathogens such as *Mycobacteria tuberculosis*, *Plasmodium* species, and arthropod-borne viruses (e.g., yellow fever virus or dengue). The hypothesis of this study is that acute MeV infection induces a loss of pre-existing immunity to pathogens prevalent in select African regions. Compared to those who do not have acute MeV infection, those with acute MeV infection will: 1) lose antibody diversity from previous infections and/or vaccinations compared to baseline; 2) have an altered immune response to a known exposure at different time-points (early vs. late) post measles infection (exposure timing is controlled by using the rabies vaccine as a controlled immune stimulus) 3) have an increased incidence of illnesses or healthcare encounters in the year following acute MeV infection.

The data gathered in this study will better characterize the effect and mechanism of measles-induced immune amnesia in children. The design of this study is described.

## Materials and methods

### Ethical statement

National Institute of Allergy and Infectious Diseases (NIAID) is the sponsor of the study. The main protocol and each of the site-specific appendices received approval from local ethics committees and/or health authorities prior to study implementation including the University of Sciences, Techniques and Technologies Bamako Mali #2023/210/CE/USTTB -Sept 18, 2023, and Guinea CNERS # 145/CNERS/23 -Sept 8, 2023. Prior to initiation of any study procedures, all study participants had an informed consent signed by a parent or guardian, and an assent as applicable, in a language they understand.

### Clinical study design

Study Overview: "Immune Amnesia Following Measles Infection in Select African Regions" (ClinicalTrials.gov Identifier: NCT06153979), is a prospective, observational, longitudinal study being conducted in West Africa to investigate the effects of MeV infection on pre-existing immunity, vaccine response, and susceptibility to subsequent illnesses. A total of 256 children are being enrolled into 1 of 2 arms: acute MeV infection (cases) or no acute MeV infection (controls). The sample size calculations are based on the primary endpoints: geometric mean RVNA titers and mean MIPSA score for the global antibody analysis and are detailed in the Statistical Aspects section. Briefly, there is 90% power to detect a standardized difference of 0.85 between cases and controls for the geometric mean RVNA titer for the groups that receive the vaccine early vs. late. There is 90% power to detect a standardized difference in the MIPSA score of 1 between cases and controls. Both of the standardized differences (0.85 and 1.00) are as large as those observed previously [26]. Acute measles is defined as signs and symptoms consistent with measles and confirmed in the laboratory using measles-specific RT-PCR on upper respiratory specimens and anti-measles IgM on blood samples. Controls are defined as an absence of signs and symptoms, negative RT-PCR, negative on anti-measles IgM, and positive on anti-measles IgG. (Controls who are otherwise eligible at screening but with a negative anti-measles IgG are vaccinated and enrolled.) Measles RNA (YouSeq RT-PCR measles, Cat no. YSL-qP-EC-Measles-100), anti-measles IgM (Abcam ELISA, Cat no. ab108751), and anti-measles IgG (Abcam ELISA, Cat no. ab108750) will be tested using the indicated commercially available kits that are verified in each laboratory. Blood samples are collected at: screening (Day 0), at an optional visit to repeat IgM serology (Day 7–10), and during follow-up visits on Day 14, Week 13, and Week 52. Plasma and, where available, PBMCs will be cryopreserved. Minimum blood volumes are collected in accordance with local IRBs. The blood samples will be used to address the primary, secondary, and exploratory objectives as outlined in Table 1. For children positive for measles, the upper respiratory sample and/or plasma from screening visit (Day 0) will be used to identify the measles genotype. In the children who test negative for measles but present with signs and symptoms consistent with measles, we will explore other viral pathogens as outlined in the exploratory objectives in Table 1.

**Table 1. Study objectives and endpoints.**

| Objectives | Endpoints | Justification For Endpoints |
|---|---|---|
| **Primary** | | |
| To determine if MeV infection induces a loss of pre-existing immunity (immune amnesia) to endemic pathogens at Week 13 after baseline in children in select African regions. | Mean change in a panel of antibody levels over 13 weeks as measured by multiplex serological methods (e.g., Molecular Indexing of Proteins by Self-Assembly (MIPSA) with an updated VirScan library and additional libraries with endemic bacteria and parasite) and targeted ELISAs for confirmation using plasma collected at Screening (Day 0) and Week 13. | These methods detect antibodies against multiple diverse pathogens and epitopes and assess both neutralizing and non-neutralizing antibodies. |
| To determine the effect of MeV infection on immune response to a controlled immune stimulus (rabies vaccination) at early and late timepoints post-infection. | Geometric mean RVNA titer 5–6 weeks after the first rabies vaccine dose using plasma from D14 (randomization) as pre-vaccine and at either Week 13 (early randomization) or Week 52 (late randomization) as post-vaccine. | These tests will measure the immune response and functionality of antibodies raised against the rabies vaccine. |
| | Proportion of subjects with an RVNA titer ≥ lower limit of quantification 5–6 weeks after the first rabies vaccine dose using plasma from D14 (randomization) as pre-vaccine and at either Week 13 (early randomization) or Week 52 (late randomization) as post-vaccine. | |
| | Proportion of subjects with rabies virus neutralizing antibodies (RVNA) titer ≥ 0.5 International Units per milliliter (IU/mL) as measured by rapid fluorescent focus inhibition test 14 days after the last PrEP regimen vaccination using plasma from D14 (randomization) as pre-vaccine and at either Week 13 (early randomization) or Week 52 (late randomization) as post-vaccine. | |
| **Secondary** | | |
| To determine if there is an increase in healthcare system encounters in the year following enrollment in children with recent MeV infection compared to those without recent MeV infection. | Mean number of non-study sick visit healthcare system encounters during the 1-year follow-up. | Visits to the healthcare system for sick visits in the year following enrollment in children with recent MeV infection will help determine the clinical impact of the immune amnesia in an objective manner. |
| To determine if MeV infection induces a loss of pre-existing immunity (immune amnesia) to endemic pathogens at Week 52 after baseline in children in select African regions. | Mean change in a panel of antibody levels over 52 weeks as measured by multiplex serological methods (e.g., MIPSA with an updated VirScan library and additional libraries with endemic bacteria and parasite) and targeted ELISAs for confirmation using plasma from screening (Day 0) and Week 52. | These methods detect antibodies against multiple diverse pathogens and epitopes. These represent the spectrum of both neutralizing and non-neutralizing antibodies. |
| **Exploratory** | | |
| To assess the appearance and continued production of B and T cells, especially ASCs from peripheral blood mononuclear cells (PBMCs) in circulation following MeV infection in children 1–15 years old in a subset of participants, depending on availability of collected blood samples. | Mean change in cell number and functionality from baseline (screening) to 14 days. PBMCs from screening (Day 0) and D14. | Assessing the number and functionality of the B and T cells including ASCs will give insight into the impact of MeV on B cell development and humoral immunity. |
| To attempt to identify the etiology of illness in children who present with signs and symptoms consistent with measles and are screened but not enrolled due to negative measles PCR and IgM. | Unbiased next generation sequencing for pathogen discovery directly from the oropharyngeal (OP)/nasopharyngeal (NP) swab and/or plasma from screening (Day 0). | This may identify other pathogens that may mimic the clinical presentation of MeV in children who have symptoms but test negative. |
| To identify the genotypes of MeV collected in the study. | Next generation sequencing or targeted Sanger sequencing directly from the OP/NP and/or plasma from screening (Day 0). | This will allow identification of different genotypes of MeV in the countries enrolling participants. |

*(Continued)*

**Table 1.** (Continued)

| Objectives | Endpoints | Justification For Endpoints |
|---|---|---|
| To characterize the phenotype of the rash and/or Koplik spots in children who present with signs and symptoms consistent with measles who test positive for measles (polymerase chain reaction [PCR] and/or IgM positive) and who test negative for measles (PCR and IgM negative). | Differences in rash patterns/presentation in children who test positive versus those who test negative for measles based on analysis of photographs. | This will allow for a comparison of rash and/or Koplik spot presentation in potential cases that end up testing positive or negative for measles in the laboratory. This will complement the exploratory objective of identifying the etiology via unbiased next general sequencing in the participants who test negative for measles. In addition, we can contribute images of measles rash on dark skinned children and Koplik spots to the medical literature. |

To explore how recent MeV infection affects the ability to respond to a new previously unencountered pathogen at two different timepoints following MeV infection, participants will receive rabies vaccine pre-exposure prophylaxis (PrEP) with Verorab inactivated rabies vaccine (Sanofi) as part of the study. PrEP is given as none of the children in the study would have previously been vaccinated or exposed to rabies. PrEP is intended to protect the recipient in case they are subsequently exposed to rabies. Previous studies have shown that children who receive a vaccine following MeV infection may have a lower antibody response to the vaccine [30]. The rabies vaccine was chosen because it is an immunogen to which most participants have not been exposed and thus can be used to measure the immune response to a newly encountered antigenic stimulus. Additionally, as rabies is estimated to cause 59,000 human deaths annually worldwide with the highest mortality in Africa [31,32], the provision of rabies PrEP is a benefit to participants in the study [33]. All children in each arm will receive rabies vaccination (3-dose series at Days 1, 7 and 28 given as PrEP), with the first dose randomized to either Week 8 or Week 47 after enrollment. WHO now recommends a 2-dose PrEP schedule [32] but we decided to use the older 3-dose schedule from the Verorab package insert (https://www.medicines.org.uk/emc/files/pil.15572.pdf) to give the acute measles cohort children the best chance of developing an immune response in the event of persisting immune amnesia. Biological samples will be collected after vaccination to assess if the immune stimulus (rabies vaccine) response differs: 1) between children with and without MeV infection, and 2) based on the timing of the receipt of the rabies vaccine. A participant who is potentially exposed to rabies during study participation will be provided rabies post exposure prophylaxis (PEP) with Verorab vaccine, with or without human rabies immune globulin, based on published recommendations [33]. The study schedule is depicted in Fig 1.

The study team will collect, to the extent possible, information on healthcare encounters that take place during the year-long follow-up. This will be compared between cases and controls to determine if there is a difference in the number of encounters by study group.

The study objectives and endpoints are shown in Table 1.

## Eligibility

The study inclusion and exclusion criteria are shown in Table 2. Children aged 1–15 years who are eligible after initial screening are enrolled in Group 1 of children with acute MeV infection (cases) or Group 2 of children without acute MeV infection (controls) eligible for the study and will be referred for care. An upper respiratory specimen (oropharyngeal (OP)/ nasopharyngeal (NP)) for MeV PCR and plasma for measles IgM/IgG serology are collected. Plasma and PBMCs are also collected at screening for research assays. PBMCs will only be collected in Mali due to operational constraints and lack of dry ice in Guinea. Anyone suspected of having measles at the time of screening will be treated with Vitamin A 200,000 IU by mouth each day for 2 days, as recommended by the WHO [34].

| Evaluation/ Procedure | Visit 0 Screening/ Enrollment[c] Blood Draw Day 0[c] | Visit 1 Randomization Blood Draw Day 14[c] (±2 days) | Visit 2[a] Early rabies vaccine Dose 1 Week 8 (±3 days) | Visit 3[a] Early rabies vaccine Dose 2 Week 9 (+2 days) | Visit 4[a] Early rabies vaccine Dose 3 Week 11-12 | Visit 5 Blood draw Healthcare Ques Week 13 (+3 days) | Visit 6 Healthcare Ques Week 26 (±2 weeks) | Visit 7 Healthcare Ques Week 39 (±2 weeks) | Visit 8[b] Late rabies vaccine Dose 1 Week 47 (±3 days) | Visit 9[b] Late rabies vaccine Dose 2 Week 48 (+2 days) | Visit 10[b] Late rabies vaccine Dose 3 Week 50-51 | Visit 11 Blood draw Healthcare Ques Week 52 (+3 days) |
|---|---|---|---|---|---|---|---|---|---|---|---|---|
| Informed consent/assent | X | | | | | | | | | | | |
| Inclusion/ Exclusion[c] | | X | | | | | | | | | | |
| Vital signs[d] | X | X | X[d] | X[d] | X[d] | X | X | X | X[d] | X[d] | X[d] | X |
| MUAC[e], height, weight | X | | | | | | | | | | | X |
| Demographics | X | | | | | | | | | | | |
| Upper respiratory specimen (OP or NP swab) for measles PCR | X | | | | | | | | | | | |
| HIV[f] | X[g] | | | | | | | | | | | X[f] |
| Hemoglobin level[h] | X | X | | | | X | | | | | | X |
| Urine pregnancy test[h] | X[i] | X | X[i] | X[i] | X[i] | X | | | X[i] | X[i] | X[i] | X |
| Measles IgM/IgG | X[g] | | | | | | | | | | | |
| Measles IgM | | X | | | | | | | | | | |
| Medical history | X | | | | | X | | | | | | X |
| Concomitant medications | X | X | X | X | X | X | X | X | X | X | X | X |
| Physical examination | X | X | | | | X | X | X | å | | | X |
| Photos (measles rashes) | X | | | | | | | | | | | |
| Randomization to early or late rabies vaccine | | X | | | | | | | | | | |
| Measles vaccine[j] | | X | | | | | | | | | | |
| AE assessment | | X | X | X | X | X | X | X | X | X | X | X |
| Healthcare Utilization Questionnaire | | | | | | X | X | X | | | | X |
| Rabies vaccine | | | (X) | (X) | (X) | | | | (X) | (X) | (X) | |
| Research blood collection[k] | X | X | | | | X | | | | | | X |
| Vitamin A | X[l] | | | | | | | | | | | |

**Fig 1. Study schedule.** Abbreviations: AE, adverse event; HIV, human immunodeficiency virus; Ig, immunoglobulin; MUAC, mid-upper arm circumference; NP, nasopharyngeal; OP, oropharyngeal; PCR, polymerase chain reaction. (X) = Each participant will receive only one 3-dose series of rabies vaccine according to randomization to early group (vaccination at Visits 2, 3, and 4) or late group (vaccination at Visits 8, 9, and 10). [a] EARLY rabies vaccination: Only participants randomized to EARLY rabies vaccine attend Visits 2, 3, 4. Visit 2 occurs 8 weeks after Day 0; Visit 3 occurs 7 days after Visit 2; Visit 4 occurs 21 to 28 days after Visit 2 AND at least 14 days after Visit 3. [b] LATE rabies vaccination: Only participants randomized to LATE rabies vaccine attend Visits 8, 9, 10. Visit 8 occurs 47 weeks after Day 0; Visit 9 occurs 7 days after Visit 8; Visit 10 occurs 21 to 28 days after Visit 8 AND at least 14 days after Visit 9. [c] Day 0 is defined as the day screening occurs and the first research blood sample is collected. Inclusion/exclusion criteria evaluation for enrollment will be completed when PCR and measles IgG/IgM and HIV results are available, generally by Day 14. All potential participants (or guardian/parent) will be informed of eligibility status by a designated study staff by phone, if possible, prior to the Day 14 scheduled study visit or at the Day 14 scheduled study visit. Ineligible participants may be invited to the site to discuss test results and referral for measles vaccination if serum IgG and IgM are both negative. [d] Vital signs include temperature, heart rate, and respiratory rate; may be obtained prior to rabies vaccine administration, if clinically indicated. [e] MUAC in children 12 to 59 months old. [f] HIV testing for children younger than 24 months will be performed via PCR and repeated with Rapid Diagnostic Test (RDT) at end of study to adhere to national guidelines; HIV RDT will be performed at screening for participants 24 months and older and will not be repeated at end of study. [g] HIV and measles serum IgM/IgG results must be reviewed for eligibility confirmation prior to Day 14. [h] Blood will be collected for hemoglobin test via finger and/or heel stick. Hemoglobin and urine pregnancy rapid test results must be reviewed prior to venous blood sample collection. [i] Females of child-bearing potential must have a negative pregnancy test prior to enrollment and each rabies vaccine administration. [j] Measles vaccination to be administered to control participants (through the study or referral to Ministry of Health vaccination program) if serum measles IgG and IgM are both negative. [k] 5 mL is the maximum volume that will be obtained at each blood draw. [l] Vitamin A will be administered at Day 0 to all participants with clinical measles per WHO guidelines: 2 oral doses of 200,000 IU given 24 hours apart. The first dose will be administered at the study site and the second will be provided to the parent to be administered 24 hours later.

## Study visits according to initial protocol

A schematic depiction of the study visit timing is shown in Fig 2. During Visit 0 on Day 0, demographic, medical history, vaccination history and concomitant medication information are collected, and a physical exam is performed. Mid upper arm circumference is obtained in children ≤59 months. Due to the turnaround time for the MeV PCR and measles IgM/IgG

**Table 2. Study inclusion and exclusion criteria.**

| Inclusion Criteria | Exclusion Criteria |
|---|---|
| 1. Aged 1–15 years.<br>2. Ability of the participant's legal or culturally acceptable representative to provide informed consent.<br>3. Ability to give assent, as appropriate.<br>4. Stated willingness of parent/guardian and participant as appropriate, to comply with all stufdy procedures.<br>5. Willingness to receive rabies vaccine.<br>6. Meet the criteria for assignment to Group 1 or Group 2, as follows:<br> a. Group 1, cases (acute MeV infection):<br> • Clinical signs and symptoms suggestive of acute MeV infection (Koplik spots or skin rash) AND<br> • Laboratory confirmed measles:<br> ◦ Upper respiratory specimen (swab) PCR for measles positive.<br> OR<br> ◦ Serum IgM for measles positive.<br> b. Group 2, controls (no acute MeV infection):<br> • No clinical signs and symptoms suggestive of acute MeV infection (Koplik spots or skin rash) AND<br> • Upper respiratory specimen (swab) PCR negative for MeV AND<br> • Serum measles IgM negative AND<br> • Serum measles IgG positive and previously vaccinated for measles (2nd dose will be offered if appropriate). If serum measles IgG is negative, participant must be willing to be vaccinated regardless of prior measles vaccine history to meet this criterion. | 1. HIV infection or any other immunosuppressive condition or medications.<br>2. Pregnant or lactating.<br>3. History of prior measles or immunologic evidence of prior measles in the absence of prior measles vaccination.<br>4. Severe anemia, defined as hemoglobin less than 8 g/dL.<br>5. Any acute or chronic condition which, in the opinion of the investigator, constitutes a contraindication to participation in this study. |

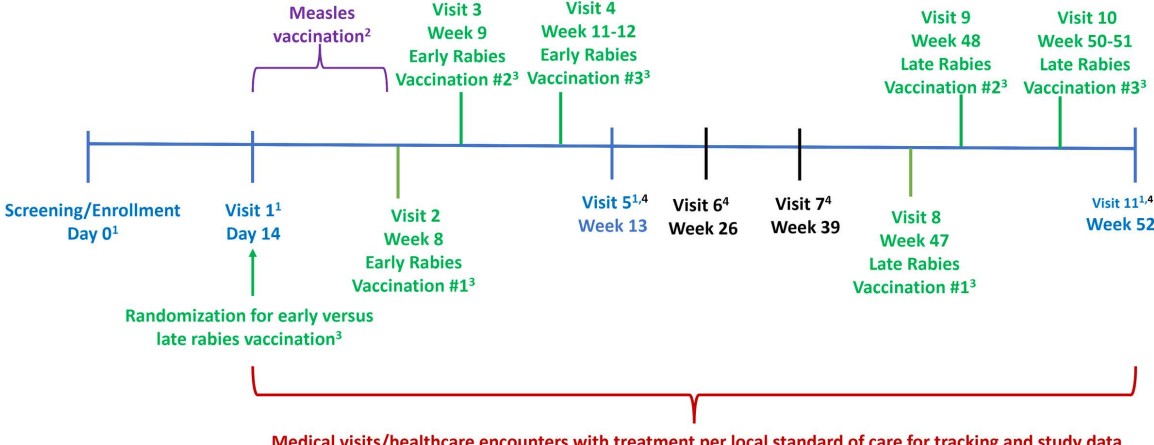

**Fig 2. Study schematic.** [1]Research blood samples. [2] Measles vaccination in controls if confirmed to have negative serum measles IgG. [3] Rabies vaccine Dose 2 is given 7 days +2 days after Dose 1, and Dose 3 is given at Day 21 to Day 28 after Dose 1 and at least 14 days after Dose 2. [4]Healthcare utilization assessment. Color coding: Blue: visits with research blood samples. Orange: optional visit with repeat for measles serology (IgM) for inconclusive results from the Day 0 test. Purple: visit with measles vaccination in controls if confirmed to have negative measles IgG. Green: All activities and visits pertaining to the rabies vaccine. Black: visits with healthcare utilization assessment. Red: Period in which the healthcare encounters will be recorded.

in the laboratory, eligibility as a case or control is confirmed at Visit 1 on Day 14. Measles vaccination will be administered to controls who do not have a positive measles IgG at this visit. Once enrolled, participants will be randomized to receive 3 doses of rabies vaccine as PrEP early (doses at Weeks 8, 9, 11–12) or late (doses at Weeks 47, 48, 50–51). At Weeks 26 and 39, a healthcare questionnaire will be administered. Participants' last visit is scheduled for Week 52. This longer follow-up period is to allow examination of antibody repertoire dynamics paired with number of healthcare encounters,

as previous studies have shown that children may remain susceptible to infections for up to 1–3 years following measles infection. Windows around study visits are relatively short so that the targeted endpoints can be captured accurately. Visits that occur out of the window periods are permitted under special circumstances, to avoid missed visits. A research blood sample is collected at Days 0 (screening) and 14(pre-vaccines), and Weeks 13 (up to 6 weeks after first early rabies vaccine dose) and 52 (up to 6 weeks after first late Rabies vaccine dose, and 1 year follow up visit); prior to collection of blood, hemoglobin is checked in all participants via a fingerstick, and a urine pregnancy test is obtained in females starting at their reproductive age. Anyone found to be pregnant is withdrawn from the study. If the hemoglobin is < 8 g/dl at any follow up blood draw visit, the participant remains in the study, but a blood sample is not collected at that time.

## Study sites

This study is being conducted in Guinea and Mali, which were chosen based on measles epidemiology, existing research infrastructure, and ongoing partnerships between their Ministries of Health, and in Mali, the Ministry of Higher Education, and the National Institute of Allergy and Infectious Diseases (NIAID) in the United States.

The Partnership for Clinical Research in Guinea (PREGUI) aims to conduct and implement a program of national and international, high-quality research on public health priorities in Guinea, and to build and develop sustainable research capacity. This study is implemented at two sites under PREGUI, both located within treatment centers for diseases with epidemic potential at referral hospitals in Forécariah and Mafèrinyah. These sites receive referrals from surrounding health centers, health posts and communities and are each served by a multidisciplinary team.

The University Clinical Research Center (UCRC) is located at the University of Sciences, Techniques, and Technologies of Bamako in Mali. Its mission is to promote clinical research and training at international standards in Mali and within the West Africa region. The UCRC offers competence in clinical research including clinical operations, laboratory diagnosis and exploration of immune function, data management, pharmacy and community engagement. The research team recruits participants from the Commune VI Reference Health Center and the Bakorobabougou Community Health Center, both in Bamako.

The first participant was enrolled in Guinea on 01/16/2024 and in Mali on 02/26/2024. The study is currently enrolling and the expected completion of enrolment is 05/31/2025 for both Mali and Guinea.

## Amendment

After screening the first 33 potential participants with suspected measles, laboratory results confirmed measles was present in only 15 (45.5%), showing that the clinical diagnosis does not always match the validated laboratory assay results. Due to concerns that true measles cases were being missed because the IgM test was possibly conducted too early, prior to development of the IgM response but after the viremic period leading to false negative results [35–38], the protocol was amended. Thus, children highly suspected of measles based on their clinical presentation but who tested negative on both PCR and IgM at day 0 may have an optional visit between days 7–10 to retest for measles IgM [35–38]. Adding a day 7–10 re-test allows more time for the IgM response to develop. Optional photography of participants being screened as potential cases was also added to the protocol. Photographs of the rash in confirmed measles cases will be used to train study staff to improve recognition of acute MeV infection in dark skinned children and serve as an educational tool for the medical community. In screened potential cases who test measles negative, these photos will be coupled with the information gained in the exploratory aim of unbiased sequencing of other pathogens to determine the phenotype of other febrile rash illnesses that may otherwise be clinically diagnosed as measles.

## Laboratory and stored specimens

The RNA is extracted from the NP/OP swabs collected in viral transport media (VTM) at screening and tested by RT-PCR for MeV. Remaining VTM samples will be stored for sequencing and genotyping of MeV. Samples from children with signs

and symptoms consistent with measles but who test negative are stored and can be examined for other viruses using pathogen discovery sequencing methods. Measles IgG and IgM are measured using an enzyme linked immunoassay on plasma samples. The screening tests used in each country (hemoglobin, pregnancy, measles serology, HIV) all underwent verification prior to study start to ensure the reproducibility and accuracy of the results. Plasma collected at baseline (Visit 0), week 13 (Visit 5) and week 52 (Visit 11) will be tested at a commercial laboratory in the United States to measure antibody binding to epitopes of select viruses, bacteria and parasites using validated multiplex serological methods (e.g., Molecular Indexing of Proteins by Self-Assembly [MIPSA] technology with an updated library and additional libraries with endemic bacteria and parasites) [26,39]. PBMCs are isolated from blood specimens from Mali, stored in liquid nitrogen, and will be analyzed for antibody secreting cells (ASC). Plasma obtained following rabies vaccination will be shipped to a commercial laboratory accredited to perform Rapid Fluorescent Foci Inhibition Test (RFFIT) to ensure neutralizing antibody (RVNA) response to the rabies vaccine and to compare cases to controls and those randomized to early vaccination versus late vaccination. Samples not used for analyses will be stored and managed electronically in the local biorepositories under appropriate temperature with constant monitoring.

## Safety reporting and monitoring

Safety reporting and monitoring is done according to national regulatory guidelines and International Council for Harmonisation (ICH). A clinician with appropriate expertise (Principal Investigator [PI]/Designee) will assess adverse event (AE) severity, according to the "Division of AIDS Table for Grading the Severity of Adult and Pediatric Adverse Events" (https://rsc.niaid.nih.gov/sites/default/files/daidsgradingcorrectedv21.pdf) and determine the seriousness, relatedness and expectedness of the event. This may be done in coordination with the sponsor, NIAID. Per study specific standard operating procedure, only AEs related to study procedures will be followed through resolution or until the local investigator judges that the event has stabilized, and no additional follow-up is required. Safety events will be tracked and submitted to the local ethics committees according to each country's requirements as described in the site-specific appendices. All AEs will be recorded on source documents.

AEs related to the research procedures and all serious adverse events (SAEs) will be recorded in the research database, except for Grade 1 or 2 AEs that are expected. SAEs and Unanticipated Problems will be reported to Sanofi Pasteur with coded data according to the terms of a confidential Clinical Trials Agreement.

## Statistical aspects

The overall sample size of the study is 256 participants, 128 enrolled as cases and 128 as controls. The sample size accounts for a 6.5% rate of dropout from the study and participants who drop out will not be replaced. The sample size is calculated to detect a difference in the RVNA geometric mean titer between the measles cases and controls who were either vaccinated at the early time point or at the late time point). If each vaccination group (early/late) contains 128 participants with 64 cases and 64 controls, there will be 90% power to detect a standardized difference in RVNA geometric mean titer of 0.85 between cases and controls under each of the vaccination conditions. There will also be 90% power to detect an interaction between group (measles/controls) and time (early/late) if the effect size is 1.19. The interaction corresponds to showing a different response between cases and controls at early and late rabies vaccine stimulus times. There is 90% power to detect standardized mean differences of 1.00 when looking for differences in the MIPSA panel of antibody responses between cases and controls in each age group. This calculation controls for multiple comparisons. This effect size was similar to those observed in the paper by Mina et al [26].

Analysis of the rabies antibody response will be performed using multivariate linear regression on the $\log_{10}$ titer values, including terms for case/control group, early/late vaccine, country and an interaction between group and timing of vaccine. Antibody responses to a panel of antigens will be compared between case/control groups using t-tests on change from baseline. Separate analyses will be performed for the 2 age groups (1–5 and 6–15 years). A Bonferroni correction will be

used to maintain a 2-sided 0.05 error rate by adjusting for the number of antibodies in the panel. A Poisson analysis will be used to compare the number of encounters with medical providers between groups.

Control participants who become infected with measles or participants from either group who become pregnant at any time during the study will be censored at the time of infection or pregnancy. A participant who has a potential rabies exposure and receives rabies PEP during the study before s/he is scheduled (randomized) to receive rabies vaccination as PrEP will be excluded from the primary analysis of the rabies vaccine but included in the immune amnesia analysis. Participants who receive at least one dose of rabies PrEP (as randomized) will be analyzed as planned. A Statistical Analysis Plan (SAP) will be prepared for the study before analyses begin.

## Study conduct, data management, operational logistics and challenges

The study leadership consists of a core team of research physicians, statisticians, laboratory research scientists, clinical research pharmacist and operational experts from the Division of Clinical Research (DCR) of NIAID and both countries. Both countries have clinical research staff and infrastructure supported by DCR. The study protocol was developed collaboratively. The country teams are responsible for submission for ethical review, continuing review reports, quality management plans, protocol deviations, and SAE reports per country regulations and as requested by NIAID. At all stages, regular teleconferences and visits are held and include study leadership, country scientific/medical laboratory, data management and clinical operations teams, and serve as a forum for information exchange on topics such as recruitment, case adjudication, visit adherence, as well as providing feedback and suggestions on study procedures. A Publication Policy has been developed to guide the drafting and review of all study-related abstracts, presentations, and manuscripts, and a Publication Committee that includes representatives from NIAID and each of the country teams will be formed. For requests for secondary use of biospecimens, the Publication Committee will be approached to approve access to biospecimens. Electronic case report forms (CRFs), core standard operating procedure (SOP) documents and manuals of operation (MOP) were developed along with site-specific documents.

The study was designed to use electronic data capture into the Clinical Data Management System (CDMS). Malian staff enter data from study-specific paper source documents. Guinean staff use direct data entry into tablets in an online mode, where the data are saved automatically to the CDMS. If the internet connection is temporarily disrupted, the data are captured and uploaded to the CDMS once service is restored. This hybrid approach allows for more flexibility across sites, reduces need for printing CRFs, and requires less transcription of data by the sites. Direct data entry also allows sites to address any queries that arise from programmed data checks at time of entry. The completed data are then reviewed by the central data management team for consistency and accuracy. Randomization is also performed electronically in the CDMS.

Specimen kits with standard barcodes are used for specimen collection. The specimens collected are stored in-country under proper conditions with constant temperature monitoring and documented in an electronic system.

A monitoring plan is used for central monitoring, to facilitate compliance with GCP, guidelines and regulations, identify key activities and specify data to be reviewed for the study. Each plan was reviewed by the study teams in Guinea and Mali and the Measles Study protocol team to ensure adherence to the protocol, laboratory, and GCP requirements. In person and virtual trainings were conducted with the study staff in each country. Refresher trainings are offered as needed.

Each country has encountered challenges with study implementation. The Guinean team is working on enhancing its ability to use electronic systems for study documents and sample management, manage a heavy workload for the available laboratory staff, and ensure participants have a good understanding of study processes. The Malian team is working on increasing the number of screened volunteers by involving more community health centers in referring suspected measles cases to the team. Also, to enhance use by community mobilizers and to sensitize community members about the study, flipbooks, a consenting tool depicting the study procedure in easily understood language with images, have been translated into Bambara, the most spoken language in Mali. Because the sites experience power losses that compromise

refrigeration, all study vaccines are stored off site where a continuous power supply is assured and transported to the sites as needed. Finally, due to the waxing and waning measles cases in both countries, enrollment is taking longer than expected.

## Conclusion

We have designed and launched an international observational study to investigate immune amnesia following MeV infection in Guinea and Mali, which have recurrent measles outbreaks. Despite challenges of collaborative protocol development, operationalization and need for multiple authorizations to import rabies and measles vaccines, the study was successfully initiated in 2024. This effort will increase understanding on how measles immune amnesia affects the response to a previously unexposed pathogen at different time points post-acute measles infection, characterize MeV genotypes circulating in these regions, and provide insights into other viral causes of maculopapular rash illnesses in children with dark skin. This study will add to the global understanding of measles from regions previously understudied, and where measles outbreaks continue to occur due to inconsistent access to vaccines.

## Supporting information

**S1 File. The Immune Amnesia Following Measles in Select African Regions clinical research protocol version 3.0 11 March 2024.**
(DOCX)

**S2 File. SPIRIT international standard checklist for the Immune Amnesia Following Measles in Select African Regions clinical research protocol.**
(DOCX)

## Acknowledgments

We thank PREGUI study team members, Centre National de Formation et de Recherche en Santé Rurale de Mafèrinyah, Centres de Santé Amélioré de Mafèrinyah, Hôpital Préfectoral de Forecariah, Community leaders in Maferyniah and Forecariah, Guinea; UCRC study team members, Centre de Santé Communautaire de Bakorobabougou, Centres de Santé de Reference Commune VI, Direction Générale de la Santé et de l'Hygiène Publique, Direction Régionale de la Santé of Bamako and Koulikoro, Community partners, Mali; Diane Griffin (deceased) and Jessica Rubens, Johns Hopkins University. We are grateful to Sanofi-Pasteur for contributions.

## Author contributions

**Conceptualization:** Karine Fouth Tchos, Renée Ridzon, Mory Cherif Haidara, Djeneba Dabitao, Esther Akpa, Daouda Camara, David Vallée, Sekou Camara, Jamila Aboulhab, Mahamadou Diakité, Bassirou Diarra, Ilo Dicko, Alyson Francis, Cécé Francis Kolié, Sally Hunsberger, Ray Y. Chen, Issa Konate, Seydou Doumbia, Abdoul Habib Beavogui, Kathryn Shaw-Saliba.

**Methodology:** Karine Fouth Tchos, Mory Cherif Haidara, Djeneba Dabitao, Esther Akpa, Daouda Camara, Sekou Camara, Mahamadou Diakité, Bassirou Diarra, Ilo Dicko, Cécé Francis Kolié, Michel Koropogui, Caeul Lim, Seydou Samaké, Sally Hunsberger, Moussa Sidibé, Ray Y. Chen, Issa Konate, Seydou Doumbia, Abdoul Habib Beavogui, Kathryn Shaw-Saliba.

**Writing – original draft:** Karine Fouth Tchos, Renée Ridzon, Kathryn Shaw-Saliba.

**Writing – review & editing:** Karine Fouth Tchos, Renée Ridzon, Mory Cherif Haidara, Djeneba Dabitao, Esther Akpa, Daouda Camara, David Vallée, Mariam Coulibaly, Sekou Camara, Jamila Aboulhab, Mahamadou Diakité, Bassirou

Diarra, Samba Diarra, Ilo Dicko, Alyson Francis, Cécé Francis Kolié, Michel Koropogui, Caeul Lim, Seydou Samaké, Sally Hunsberger, Moussa Sidibé, Ray Y. Chen, Issa Konate, Abdoul Habib Beavogui, Kathryn Shaw-Saliba.

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
