## [Decision Letter · Decision Letter 0]

20 Dec 2024

Dear Dr. Fouth Tchos,

If applicable, we recommend that you deposit your laboratory protocols in protocols.io to enhance the reproducibility of your results. Protocols.io assigns your protocol its own identifier (DOI) so that it can be cited independently in the future. For instructions see: https://journals.plos.org/plosone/s/submission-guidelines#loc-laboratory-protocols . Additionally, PLOS ONE offers an option for publishing peer-reviewed Lab Protocol articles, which describe protocols hosted on protocols.io. Read more information on sharing protocols at https://plos.org/protocols?utm_medium=editorial-email&utm_source=authorletters&utm_campaign=protocols.

We look forward to receiving your revised manuscript.

Kind regards,

Pierre Roques, Ph.D.

Academic Editor

PLOS ONE

Journal Requirements: When submitting your revision, we need you to address these additional requirements. 1. Please ensure that your manuscript meets PLOS ONE's style requirements, including those for file naming. The PLOS ONE style templates can be found at https://journals.plos.org/plosone/s/file?id=wjVg/PLOSOne_formatting_sample_main_body.pdf and https://journals.plos.org/plosone/s/file?id=ba62/PLOSOne_formatting_sample_title_authors_affiliations.pdf 2. Please include a complete copy of PLOS’ questionnaire on inclusivity in global research in your revised manuscript. Our policy for research in this area aims to improve transparency in the reporting of research performed outside of researchers’ own country or community. The policy applies to researchers who have travelled to a different country to conduct research, research with Indigenous populations or their lands, and research on cultural artefacts. The questionnaire can also be requested at the journal’s discretion for any other submissions, even if these conditions are not met.  Please find more information on the policy and a link to download a blank copy of the questionnaire here: https://journals.plos.org/plosone/s/best-practices-in-research-reporting. Please upload a completed version of your questionnaire as Supporting Information when you resubmit your manuscript. 3. We note that the grant information you provided in the ‘Funding Information’ and ‘Financial Disclosure’ sections do not match.  When you resubmit, please ensure that you provide the correct grant numbers for the awards you received for your study in the ‘Funding Information’ section. 4. Please note that in order to use the direct billing option the corresponding author must be affiliated with the chosen institute. Please either amend your manuscript to change the affiliation or corresponding author, or email us at plosone@plos.org with a request to remove this option. 5. Please ensure that you refer to Figure 2 in your text as, if accepted, production will need this reference to link the reader to the figure. 6. Please include captions for your Supporting Information files at the end of your manuscript, and update any in-text citations to match accordingly. Please see our Supporting Information guidelines for more information: http://journals.plos.org/plosone/s/supporting-information. 7. Please review your reference list to ensure that it is complete and correct. If you have cited papers that have been retracted, please include the rationale for doing so in the manuscript text, or remove these references and replace them with relevant current references. Any changes to the reference list should be mentioned in the rebuttal letter that accompanies your revised manuscript. If you need to cite a retracted article, indicate the article’s retracted status in the References list and also include a citation and full reference for the retraction notice.

Reviewers' comments:

Reviewer's Responses to Questions

**Comments to the Author**

1. Does the manuscript provide a valid rationale for the proposed study, with clearly identified and justified research questions?

Reviewer #1: Partly

2. Is the protocol technically sound and planned in a manner that will lead to a meaningful outcome and allow testing the stated hypotheses?

Reviewer #1: Partly

3. Is the methodology feasible and described in sufficient detail to allow the work to be replicable?

Reviewer #1: No

4. Have the authors described where all data underlying the findings will be made available when the study is complete?

Reviewer #1: Yes

5. Is the manuscript presented in an intelligible fashion and written in standard English?

Reviewer #1: No

You may also provide optional suggestions and comments to authors that they might find helpful in planning their study.

Reviewer #1: This is a study protocol that will be useful in research, but important points need to be clarified, such as the choice of study cohort, key questions about the vaccination status of the children in the cohort, and the fact that the latest measles vaccination coverage figures are not very high, which should be taken into account.

It should be made clear why the rabies vaccine was used and not another vaccine. The key points are not easy to find in the text, as the sentences are a little too long and therefore difficult to understand. This protocol needs to be read by a native English speaker.

**Do you want your identity to be public for this peer review?** For information about this choice, including consent withdrawal, please see our Privacy Policy

Reviewer #1: No

---

## [Author Response · Author response to Decision Letter 1]

11 Feb 2025

"Investigating immune amnesia after measles virus infection in two West African countries: A study protocol"

This manuscript presents an important study that could significantly contribute to our understanding of immune amnesia associated with measles infection. Addressing the points mentioned above will enhance clarity, transparency, and rigor, ultimately supporting the validity of the research findings.

I'll be looking at clarity, structure, grammar and relevance of content. This article needs to be read by a native English speaker.

This manuscript was read by all of the authors which includes five native English speakers and all of the non-native English-speaking authors are proficient in English. We would like to suggest to the Editor and reviewer to refrain from using the term “native English speaker” as it makes unsupported assumptions on the authors’ origins and capacity. We appreciate the comment to improve the clarity and grammar of the manuscript and have done so in this resubmission.

Title:

The title is informative and precise. However, it could benefit from simplification, for example:

- “Study of immune amnesia following measles virus infection in West Africa”.

Response: Thank you for this consideration. The title is derived from the protocol title and what is captured in ClinicalTrials.gov Identifier: NCT06153979. We would like to propose therefore keeping the full title. The short title has been updated to reflect the suggestion from the reviewer.

Abstract:

What is the working hypothesis?

Response: The working hypothesis is that acute MeV infection induces a loss of pre-existing immunity to pathogens prevalent in select African regions. Compared to those who do not have acute MeV infection, acute MeV infection may result in 1) A loss of antibody diversity from previous infections and/or vaccinations compared to screening; 2) Altered immune response to a controlled immune stimulus (rabies vaccination) given at two different timepoints; 3) Increased incidence of illnesses or healthcare encounters in the year following acute MeV infection. The abstract has been reworked to more clearly highlight the objectives of the study.

Line 16: “Investigation of Immune Amnesia Following Measles Infection in Select African Regions.”

Correction: Make sure this sentence aligns with the article title.

Response: Line 16 aligns with the article title and reflects the protocol title as identified in ClinicalTrials.gov Identifier: NCT06153979.

Line17-18: “a prospective, observational, longitudinal study being conducted in two West African countries, Guinea, and Mali.”

Correction: Delete the comma after “Guinea”.

Response: This has been updated.

Material and methods

Line 86: Although you mention enrolling 256 children, further justification of this sample size in terms of statistical power and expected effect sizes would strengthen the manuscript. Please explain the basis for this number and if any preliminary data guided this decision.

The manuscript includes the detailed explanation for the sample size justification in the Statistical Aspect section (clean version line 295). However, for clarity and in response to the feedback, the following has been added to the material and methods (clean version line 92): “The sample size calculations are based on the primary endpoints: geometric mean RVNA titers and mean MIPSA score for the global antibody analysis and are detailed in the Statistical Aspects section. Briefly, there is 90% power to detect a standardized difference of 0.85 between cases and controls for the geometric mean RVNA titer for the groups that receive the vaccine early vs. late. There is 90% power to detect a standardized difference in the MIPSA score of 1 between cases and controls. Both of the standardized differences (0.85 and 1.00) are as large those observed previously 1.” The sample size calculations were based on the previous study that identified the loss of pre-existing antibodies (immune amnesia), which is properly referenced throughout the manuscript: Mina, M.J., et al., Measles virus infection diminishes preexisting antibodies that offer protection from other pathogens. Science, 2019. 366(6465): p. 599-606.

Line 87: “Acute MeV is confirmed by RT-PCR testing on upper respiratory specimens or IgM detection on blood samples at screening.”

Correction: Clarify “screening” by specifying the time of evaluation.

Response: We have added Day 0 (D0) to “screening” here and throughout the manuscript for clarity.

Please cite the reference of the method you will use. Please indicate which kit to use and the supplier.

Response: The following has been added (clean version line 103): “Measles RNA (YouSeq RT-PCR measles, Cat no. YSL-qP-EC-Measles-100), anti-measles IgM (Abcam ELISA, Cat no. ab108751), and anti-measles IgG (Abcam ELISA, Cat no. ab108750)” will be tested using the indicated commercially available kits that are verified in each laboratory.”

Line 88-89: “Blood samples are collected at multiple time points at screening (Day 0), at an optional visit to repeat IgM serology for inconclusive or negative Day 0 results (Day 7-10), and during follow-up visits on Day 14, Week 13, and Week 52.”

Comment: Clarify the purpose of the blood drive and what will be measured.

Response: We have clarified both in the text in the methods and Table 1 what objectives will be addressed with the blood that is collected.

The following information has been added to the text (clean version line 107): “Plasma and, where available, PBMCs will be cryopreserved. Minimum blood volumes are collected in accordance with local IRBs. The blood samples will be used to address the primary, secondary, and exploratory objectives as outlined in Table 1. For children positive for measles, the upper respiratory sample and/or plasma from screening visit (Day 0) will be used to identify the measles genotype in the children who test negative for measles but present with signs and symptoms consistent with measles, we will explore other viral pathogens as outlined in the exploratory objectives in Table 1.as outlined in the exploratory objectives in Table 1.”

Updated Table 1:

OBJECTIVES ENDPOINTS JUSTIFICATION FOR ENDPOINTS

Primary

To determine if MeV infection induces a loss of pre-existing immunity (immune amnesia) to endemic pathogens at Week 13 after baseline in children in select African regions. Mean change in a panel of antibody levels over 13 weeks as measured by multiplex serological methods (e.g., Molecular Indexing of Proteins by Self-Assembly (MIPSA) with an updated VirScan library and additional libraries with endemic bacteria and parasite) and targeted ELISAs for confirmation using plasma collected at Screening (D0) and Week 13. These methods detect antibodies against multiple diverse pathogens and epitopes and assess both neutralizing and non-neutralizing antibodies.

To determine the effect of MeV infection on immune response to a controlled immune stimulus (rabies vaccination) at early and late timepoints post-infection.

Geometric mean RVNA titer 5-6 weeks after the first rabies vaccine dose using plasma from D14 (randomization) as pre-vaccine and at either Week 13 (early randomization) or Week 52 (late randomization) as post-vaccine. These tests will measure the immune response and functionality of antibodies raised against the rabies vaccine.

Proportion of subjects with an RVNA titer ≥ lower limit of quantification 5-6 weeks after the first rabies vaccine dose using plasma from D14 (randomization) as pre-vaccine and at either Week 13 (early randomization) or Week 52 (late randomization) as post-vaccine.

Proportion of subjects with rabies virus neutralizing antibodies (RVNA) titer ≥ 0.5 International Units per milliliter (IU/mL) as measured by rapid fluorescent focus inhibition test 14 days after the last PrEP regimen vaccination using plasma from D14 (randomization) as pre-vaccine and at either Week 13 (early randomization) or Week 52 (late randomization) as post-vaccine.

Secondary

To determine if there is an increase in healthcare system encounters in the year following enrollment in children with recent MeV infection compared to those without recent MeV infection. Mean number of non-study sick visit healthcare system encounters during the 1-year follow-up. Visits to the healthcare system for sick visits in the year following enrollment in children with recent MeV infection will help determine the clinical impact of the immune amnesia in an objective manner.

To determine if MeV infection induces a loss of pre-existing immunity (immune amnesia) to endemic pathogens at Week 52 after baseline in children in select African regions. Mean change in a panel of antibody levels over 52 weeks as measured by multiplex serological methods (e.g., MIPSA with an updated VirScan library and additional libraries with endemic bacteria and parasite) and targeted ELISAs for confirmation using plasma from screening (D0) and Week 52. These methods detect antibodies against multiple diverse pathogens and epitopes. These represent the spectrum of both neutralizing and non-neutralizing antibodies.

Exploratory

To assess the appearance and continued production of B and T cells, especially ASCs from peripheral blood mononuclear cells (PBMCs) in circulation following MeV infection in children 1-15 years old in a subset of participants, depending on availability of collected blood samples. Mean change in cell number and functionality from baseline (screening) to 14 days. PBMCs from screening (D0) and D14.

Assessing the number and functionality of the B and T cells including ASCs will give insight into the impact of MeV on B cell development and humoral immunity.

To attempt to identify the etiology of illness in children who present with signs and symptoms consistent with measles and are screened but not enrolled due to negative measles PCR and IgM. Unbiased next generation sequencing for pathogen discovery directly from the oropharyngeal (OP)/nasopharyngeal (NP) swab and/or plasma from screening (D0). This may identify other pathogens that may mimic the clinical presentation of MeV in children who have symptoms but test negative.

To identify the genotypes of MeV collected in the study. Next generation sequencing or targeted Sanger sequencing directly from the OP/NP and/or plasma from screening (D0). This will allow identification of different genotypes of MeV in the countries enrolling participants.

To characterize the phenotype of the rash and/or Koplik spots in children who present with signs and symptoms consistent with measles who test positive for measles (polymerase chain reaction [PCR] and/or IgM positive) and who test negative for measles (PCR and IgM negative).

Differences in rash patterns/presentation in children who test positive versus those who test negative for measles based on analysis of photographs. This will allow for a comparison of rash and/or Koplik spot presentation in potential cases that end up testing positive or negative for measles in the laboratory. This will complement the exploratory objective of identifying the etiology via unbiased next general sequencing in the participants who test negative for measles. In addition, we can contribute images of measles rash on dark skinned children and Koplik spots to the medical literature.

I didn't understand the reason for week 52, could I have an explanation?

Response: Week 52 was chosen for a few reasons. First, participants are randomized to receive rabies at an early timepoint (8 weeks) or a late timepoint (47 weeks) following their measles infection. In consultation with Sanofi, the peak anti-rabies antibody production occurs at 5-6 weeks after the first dose of the vaccine, which resulted in 52 weeks. Second, the Mina, M.J., et al., Science, 2019. 366(6465): p. 599-606 paper looking at the global loss of pre-existing antibodies looked at the change in antibody binding profile at a median of 10 weeks post-infection. The primary objective will look at 13 weeks at this study. We also wanted to follow the children up to one year to examine a longer follow-up period and potential return of the antibody repertoire paired with the number of healthcare encounters. Previous studies have shown that children remain susceptible to other infections up to 1-3 years following infection [Behrens L, et al. Pediatr Infect Dis J 2020; 39(6): 478-82]. We have clarified this in the study visits section (clean version line 198)

Suggestion: Break down this sentence for greater clarity.

Response: The sentence has been changed from “Blood samples are collected at multiple time points at screening (Day 0), at an optional visit to repeat IgM serology for inconclusive or negative Day 0 results (Day 7-10), and during follow-up visits on Day 14, Week 13, and Week 52.”

To

“Blood samples are collected at: screening (Day 0), at an optional visit to repeat IgM serology (Day 7-10), and during follow-up visits on Day 14, Week 13, and Week 52.”

Line 90-93: “These blood samples will be tested to evaluate both humoral and cellular immune responses to endemic pathogens to measure variations in antibody diversity and antibody secreting cells (ASCs).”

Correction: Make sure the term “humoral” is understood in context.?

Response: This sentence has been removed and instead replaced by the following to clarify the intended use of plasma and PMBCS samples obtained from blood collection:

“For children positive for measles, the upper respiratory sample and/or plasma from screening visit (Day 0) will be used to identify the measles genotype In the children who test negative for measles but present with signs and symptoms consistent with measles, we will explore other viral pathogens as outlined in the exploratory objectives in Table 1.”

The objectives in Table 1 have also been updated accordingly.

You based your assessment of loss of immunity in children with measles on the fact of acute infection. Have you taken into account whether or not they have received one or 2 doses of measles vaccine? Because I think the length of time, they've been vaccinated can influence the immune response.

Response: Thank you for this consideration. The reviewer is correct in that we based our assessment on the loss of immunity in children with acute measles infection, regardless of their vaccine history. We do not think the measles vaccine regimen will have a measurable effect in our assessment of immunity loss, but we are collecting necessary data to examine this and can incorporate this into the analysis. As part of the case report forms and discussion with the parents, we are collecting measles vaccination history as reported by the parents and/or including information from the child’s vaccination record. The vaccination record is standard in Guinea and Mali. Both from the reported history and from the number of doses (1 or 2) are recorded in the case report form. Regardless of the vaccination timing and number of doses, children both in the cases and controls are also tested for anti-measles IgG at screening (D0).

Line 94-96: “To explore how recent MeV infection may affect the child’s ability to respond to a controlled immune stimulus, all participants will receive rabies vaccine pre-exposure prophylaxis (PrEP) using Verorab inactivated rabies vaccine.”

Comment: The use of “pre-exposure prophylaxis” could be further explained for readers less familiar with the term.

Response: PrEP is a common term for giving a vaccine or drug prior to exposure. We have further clarified by adding in clean version line 118: “PrEP is given as none of the children in the study would have previously been vaccinated or exposed to rabies. PrEP is intended to protect the recipient in case they are subsequently exposed to rabies.”

Figure 2: I didn't understand the color differences. Please be more explicit.

Response: Thank you for this comment. The following information has been added to the legend:

Blue: visits with research blood samples

Orange: optional visit with repeat for measles serology (IgM) for inconclusive results from the D0 test

Purple: visit with measles vaccination in controls if confirmed to have negative measles IgG

Green: All activities and visits pertaining to the rabies v

---

## [Decision Letter · Decision Letter 1]

25 Apr 2025

Investigating immune amnesia after measles virus infection in two West African countries: A study protocol

PONE-D-24-50742R1

Dear Dr. Karine G Fouth Tchos

We’re pleased to inform you that your manuscript has been judged scientifically suitable for publication and will be formally accepted for publication once it meets all outstanding technical requirements.

Kind regards,

Javier Antonio Benavides-Montaño

Academic Editor

PLOS ONE

Additional Editor Comments (optional):

Reviewers' comments:

Reviewer's Responses to Questions

**Comments to the Author**

1. Does the manuscript provide a valid rationale for the proposed study, with clearly identified and justified research questions?

Reviewer #1: Yes

2. Is the protocol technically sound and planned in a manner that will lead to a meaningful outcome and allow testing the stated hypotheses?

Reviewer #1: Yes

3. Is the methodology feasible and described in sufficient detail to allow the work to be replicable?

Reviewer #1: Yes

4. Have the authors described where all data underlying the findings will be made available when the study is complete?

Reviewer #1: Yes

5. Is the manuscript presented in an intelligible fashion and written in standard English?

Reviewer #1: Yes

You may also provide optional suggestions and comments to authors that they might find helpful in planning their study.

Reviewer #1: Je n’ai rien d’autre à ajouter,

Ils ont pris en compte mes points de vue et ont répondu à mes attentes.

**Do you want your identity to be public for this peer review?** For information about this choice, including consent withdrawal, please see our Privacy Policy

Reviewer #1: No

---

## [Editor Report · Acceptance letter]

PONE-D-24-50742R1

PLOS ONE

Dear Dr. Hunsberger,

I'm pleased to inform you that your manuscript has been deemed suitable for publication in PLOS ONE. Congratulations! Your manuscript is now being handed over to our production team.

Kind regards,

on behalf of

Ph.D Javier Antonio Benavides-Montaño

Academic Editor

PLOS ONE